# Unexpected Negative Performance of PdRhNi Electrocatalysts toward Ethanol Oxidation Reaction

**DOI:** 10.3390/mi14050957

**Published:** 2023-04-27

**Authors:** Ahmed ElSheikh, James McGregor

**Affiliations:** 1Department of Chemical & Biological Engineering, University of Sheffield, Sheffield S1 3JD, UK; 2Mechanical Engineering Department, Faculty of Engineering, South Valley University, Qena 83523, Egypt

**Keywords:** ethanol oxidation reaction, trimetallic nanoparticles, Rh effect, surface metal segregation

## Abstract

Direct ethanol fuel cells (DEFCs) need newly designed novel affordable catalysts for commercialization. Additionally, unlike bimetallic systems, trimetallic catalytic systems are not extensively investigated in terms of their catalytic potential toward redox reactions in fuel cells. Furthermore, the Rh potential to break the ethanol rigid C-C bond at low applied potentials, and therefore enhance the DEFC efficiency and CO_2_ yield, is controversial amongst researchers. In this work, two PdRhNi/C, Pd/C, Rh/C and Ni/C electrocatalysts are synthesized via a one-step impregnation process at ambient pressure and temperature. The catalysts are then applied for ethanol electrooxidation reaction (EOR). Electrochemical evaluation is performed using cyclic voltammetry (CV) and chronoamperometry (CA). Physiochemical characterization is pursued using X-ray diffraction (XRD), transmission electron microscope (TEM), energy-dispersive X-ray spectroscopy (EDX) and X-ray photoelectron spectroscopy (XPS). Unlike Pd/C, the prepared Rh/C and Ni/C do not show any activity for (EOR). The followed protocol produces alloyed dispersed PdRhNi nanoparticles of 3 nm in size. However, the PdRhNi/C samples underperform the monometallic Pd/C, even though the Ni or Rh individual addition to it enhances its activity, as reported in the literature herein. The exact reasons for the low PdRhNi performance are not fully understood. However, a reasonable reference can be given about the lower Pd surface coverage on both PdRhNi samples according to the XPS and EDX results. Furthermore, adding both Rh and Ni to Pd exercises compressive strain on the Pd lattice, noted by the PdRhNi XRD peak shift to higher angles.

## 1. Introduction

Low-molecule alcohols such as methanol (CH_3_OH) and ethanol (C_2_H_5_OH) are appealing options for the scalable applications of fuel cells either to be reformed into hydrogen that is fed into the fuel cell or used directly for the fuel cell’s anodic oxidation [1]. Such alcohols, especially ethanol, are receiving more attention because they are considered renewable, since they can be produced from plant-based byproducts. Direct alcohol fuel cells (DAFCs) emerge as one of the fast-developing technologies to generate ecofriendly electricity and help mitigate climate change. Feeding the fuel cell anode directly with high alcohol liquid such as bioethanol presents several benefits, such as the ease of fuel storage and transport. Furthermore, ethanol is ecofriendly because it is produced from biomass. It is also energetic and hydrogen-rich. However, certain challenges against their community-scale adoption remain unresolved. One of them is the need for platinum (Pt)-based catalysts and the short Pt catalytic lifetime due to its low-tolerance towards poisoning species [2,3]. Furthermore, the catalytic systems always have two constituents: the active phase (e.g., Pt) to drive the redox reactions and a substrate or support to provide texture for the active phase. Conventionally, carbon nanomaterials (Vulcan carbon, carbon nanofibers, carbon nanotubes, graphene and others) could perform the support role due to their electronic conductivity, high specific surface area and porous structure. Some metal oxides could also play the same role and sometimes the support is hybrid carbon–metal oxide. However, Halim et al. [4] recently reviewed the application of conducting polymers as catalyst support for fuel cell electrocatalysis application. Due to their π-conjugated backbones and highly porous structures, conducting polymers could exercise a significant impact on the dispersion of catalytic metal nanoparticles.

Furthermore, direct ethanol application in fuel cells has its own limitations, since its full oxidation to CO_2_ is not easily accomplished due to the rigid C-C bond in its molecules, which favors the production of acetic acid releasing 4 electrons instead of 12. This means that two-thirds of the faradaic efficiency are lost. Yaqoob et al. [5] have extensively reviewed the Pt-based catalyst application in DEFCs. Pt-based nanocubes, nanorods and nanoflowers are emerging structural examples. Additionally, the modification of Pt by adding metal oxides such as Fe_2_O_3_, TiO_2_, SnO_2_, MnO, Cu_2_O and ZnO is an interesting research direction. According to Yang et al. [3], Pt-based catalysts are usually unable to break the C-C bond in ethanol molecules over the long-term DEFC operation. In the same paper, the authors argued that it is essential to reduce the catalyst cost to commercialize DEFC and that this might be achieved via adopting a core@shell structure while the core is of abundant material, while the shell is a noble metal. They have also argued that alloying two non-noble metals to form a core maximizes the catalytic benefits. Unlike Pt bimetallic catalysts, trimetallic Pt-based alcohol oxidation catalysts are rarely reported due to their complex reduction kinetics, especially since the three metals are likely to have similar atomic radii, electronic structures and redox potentials [6].

Palladium (Pd) possesses a Pt-like physical structure and catalytic properties, but it is more abundant [7,8]. Furthermore, Pd enjoys a better tolerance towards the carbonaceous species that might attack its active sites [9]. The active metal component is usually deposited on a high-surface area conducting material such as carbon black to block the nanoparticle coalescence [10]. To further reduce the noble-Pd utilization and enhance its catalytic performance, a second transition metal can be added to Pd to produce a supported a binary metal catalyst system. The successful bimetallic systems for alcohol oxidation include PdAu [11], PdCo [12], PdNi [13,14], PdCu [15] and PdRu [16,17]. According to these works, the second metal activates the bifunctional mechanism through low-potential generation of oxygen species. Additionally, it modifies the geometry and electronic configuration of Pd. Three-dimensional architectures of Pd, Pd-Au and Pd-Ir aerogel were quickly synthesized by the self-assembly method and have shown enhanced EOR performance compared with the conventional Pd/C [18,19,20,21]. Ni is extensively reported to have a positive impact on Pd alcohol oxidation performance [13,14,22,23]. The gist of these works is that adding Ni to Pd can decrease its particle size, activate water and tune in the Pd electronic configuration. Adding Rh has enhanced the Pd ethanol oxidation performance [7,24], while the opposite was reported by Fontes et al. [25]. Furthermore, according to [26,27,28], Rh can, on its own, split the C-C bond in ethanol molecules and significantly enhance the CO_2_ selectivity of EOR, even though its activity is much lower than its Pd and Pt counterparts. This point, however, is a point of contention, since other research efforts have found the opposite results [25,29].

Heightening the impact of a second cocatalyst metal by adding a third metal is at the beginning of laboratory investigation [7,30,31,32,33,34,35]. In this work, trimetallic C-supported and Pd-based catalysts (with Rh and Ni added) are synthesized, characterized and applied for alkaline ethanol oxidation. It is hypothesized that since the individual addition of Ni or Rh enhances the Pd/C EOR kinetics, the simultaneous addition of Rh and Ni can maximize the catalytic benefits of Pd ethanol oxidation performance.

## 2. Experimental Work

### 2.1. Chemicals

Vulcan carbon XC72R was procured from Cabot Corp (Boston, MA, USA). Metal salts of PdCl_2_ (99 wt.%), AgNO_3_ (97 wt.%) and RhCl_3_ (99% wt.%) were purchased from Sigma-Aldrich (St. Louis, MO, USA). The solid metal precursors were converted into liquid solutions and stored for frequent use. NABH_4_ (98 wt.%), KBr (95% wt.%), ethanol (100% absolute), KOH (85 wt.%) and 2-propanol (99%) were also purchased from Sigma-Aldrich (St. Louis, MO, USA). These chemicals were used as received.

### 2.2. Catalyst Synthesis

The electrocatalysts were prepared using a quick borohydride reduction following the works of [36,37,38] with a little change. Two trimetallic samples were prepared and identified as PdRhNi/C and Pd_4_Rh_2_Ni_1_/C following their associated Pd:Rh:Ni nominal atomic ratios of 1:1:1 and 4:2:1. Monometallic samples of Pd/C, Rh/C, and Ni/C were prepared for comparison purposes. The catalyst metal loading was fixed at 12 wt.% for all samples. Table 1 demonstrates the stoichiometric added quantities of carbon and metal precursors to prepare each catalyst. The metal precursors were added into a 100 mL mixture of deionized water and 2-propoanol (50/50:*v*/*v*). Prior to that, KBr was added to work as a capping agent and the KBr/Me atomic ratio was 1.5. Then, Vulcan carbon (132 mg each sample) was added into the mixture and sonicated for 30 min. Thereafter, the mixture was transferred to a magnetic stirrer plate. Following this, the NaBH_4_ solution (40 mL, 0.2 M) was added to the mixture in one portion. The whole mixture was kept under stirring for 30 min, and after that it was left for 24 h for particle precipitation. The mixture was then washed with deionized water and acetone under vacuum filtration. Then, the wet catalyst powder was dried at 120 °C in a vacuum oven overnight.

### 2.3. Catalyst Evaluation

The catalysts were applied for alkaline ethanol oxidation reaction (EOR). The catalytic performance was evaluated by means of cyclic voltammetry (CV) and chronoamperometry (CA). These tests were undertaken in a magnetically stirred 3-electrode cell containing either 1 M KOH or 1 M (KOH + C_2_H_5_OH) solutions. The working electrode was a mirror-shining glassy carbon (GCE, Φ3 mm) one painted with the respective catalyst slurry. The counter and reference electrodes were a Pt wire and Ag/AgCl (sat ’KCl), respectively. To prepare the catalyst slurry, 5 mg of the respective catalyst powder was sonicated in a mixture of 2 mL of ethanol and 25 µL of Nafion ^®^117 (5 wt.%) for 1 h. A total of 5 µL of the slurry was micro-pipetted on the GCE electrode and left a few minutes to dry; this step was repeated 5 times. The total metal loading on the working electrode for all samples was 85 µg/cm^2^. The CV scan rate was 50 mV/s, and the 20th cycle was selected for analysis. CA was performed at −0.3 V and +0.1 V vs. NHE.

### 2.4. Catalyst Characterization

X-ray diffraction (XRD) was applied to investigate the catalyst structure using a Bruker D2 Phaser (Bruker, Billerica, MA, USA) equipped with Cu Kα radiation at 30 kV and 10 mA. The scan rate applied was 12°/min. The chemical composition was investigated using an energy-dispersive X-ray spectroscopy (EDX) detector attached to a Jeol JSM 6010LA SEM microscope (Jeol, Tokyo, Japan) operating at 20 kV. The catalyst morphology was analyzed using a Philips C100 TEM microscope (Philips/FEI Corporation, Eindhoven, The Netherlands) with a LaB6 filament operating at 100 kV. X-ray photoelectron spectroscopy (XPS) was performed to analyze surface chemistry. The equipment used was a Thermo-Scientific K-Alpha^+^ (Waltham, MA, USA) with an aluminum X-ray source (72 W). The energy step and step size (to record high-resolution peaks) were 50 eV, 0.1 eV. Argon ions were used to neutralize the charge and CasaXPS (Shirley background and Scofield cross sections) was used to analyze the data with −0.6 energy dependence.

## 3. Results and Discussion

The raw research data of this work are available at DOI: 10.17632/hp22gbrn9d.1.

### 3.1. Electrochemical Characterization

Figure 1A demonstrates the CV scan results in 1 M KOH at 50 mV/s in a stirred three-electrode half-cell in which the working electrode was Pd/C, Ni/C, Rh/C, PdRhNi/C or Pd_4_Rh_2_Ni_1_/C ink deposited on GCE. The obtained current was divided by the total metal load in the respective catalyst. The Rh voltammogram showed a low potential hydrogen adsorption/absorption peak around −600 mV vs. NHE. A similar peak was also noted with both PdRhNi samples. A little oxidation of Rh occurred towards the forward scan end that was reversed in the backward scan around −100 mV. Similar Rh voltammograms were also reported [1,39]. The Ni one did not show a significant activity at low potential (below +500 mV). Above that, a steep rise in the current density was noted until the forward sweep end, and it was reversed in the backward scan at slightly smaller potential. This was due to the oxidation of NiOOH to Ni(OH)_2_. A minor Ni reduction peak was noted at a much lower potential (−100 mV), which is suggestive of multiple Ni oxidation states’ presence on the catalyst surface. A comparable current rise was noted with the PdRhNi sample in the forward and backward scans. Unlike Rh, the Pd voltammogram did not show a H_ads/abs_ peak around −600 mV, and instead, a OH^−^ adsorption peak was seen around −300 mV. Above 0 mV, the PdO formation occurred incrementally with the applied potential. The PdO was reduced back in the reverse scan around −150 mV. The trimetallic voltammograms shared the features of the three metals, which were Rh H_ads/abs_, Pd oxidation/reduction and Ni oxidation/reduction following their atomic concentrations.

After adding ethanol (Figure 1B), significant changes in the catalytic voltammograms could be seen. For instance, the Rh and Ni samples had a zero-like activity towards EOR. On the other hand, with Pd, the ethanol adsorption at low potential (around −500 mV) suppressed the H adsorption. With increasing the potential and the growing adsorbed OH^−^ species, the adsorbed ethoxy species was oxidized and removed from the Pd active site, increasing the drawn current. This trend continued until around +50 mV, after which the current declined due to the gradual reduction of Pd active sites. In the backward scan, the PdO was reduced back, as shown in Figure 1A, so there is a sharp current rise in Figure 1B around −100 mV due to the adsorption of fresh ethanol species. Little backward shoulder peaks appeared probably due to the stirring conditions of the electrolyte. The Pd mass activity of the trimetallic samples, surprisingly, was lower than their monometallic Pd/C counterpart. Furthermore, there was a forward peak-overlap at approximately +50 mV and +400 mV on both trimetallic samples, with the 400-mV one more intense than the 50-mV one on the PdRhNi/C one, while the opposite was true for Pd_4_Rh_2_Ni_1_/C. The PdRhNi/C voltammogram shows an intense Ni oxidation/reduction around 600 mV. The effect of Ni added to Pd/C has been repeatedly been reported as beneficial for ethanol oxidation reaction (EOR) [14,40,41]. However, there is ambiguity about the impact Rh could exercise on EOR. For instance, Fontes et al. [25] found that, similar to the current work, PdRh/C underperformed Pd/C on EOR. On the contrary, Almeida et al. [7] prepared PdNi/C and PdRhNi/C—with a core@shell structure—that outperformed Pd/C on EOR.

Figure 2 shows the two-step CA scan results at −300 mV and +100 mV vs. NHE for 0.5 h each. These potentials were picked up because they represent the OH adsorption (−300 mV) and Pd surface oxidation (+100 mV), according to Figure 1A. The current drawn at −300 mV is generally lower than that at +100 mV, which could be attributed to the EOR enhancement via increased oxygen species adsorption at higher applied potentials. The Pd/C drawn current is higher than both trimetallic samples, suggesting the monometallic sample is more tolerant than CO poisoning species since CO-like species are anticipated during EOR. Such species adsorb strongly and block the Pd active sites from further EOR, even though their adsorption on Pt is much stronger. At +100 mV, the difference between monometallic Pd/C and trimetallic ones is significantly in favor of Pd/C. At such high overpotential (+100 mV), Pd active sites are quickly lost, making the fast current decay, even though they are still higher than the current drawn from both trimetallic samples.

### 3.2. Physiochemical Characterization

Figure 3 shows the XRD patterns of Pd/C, Ni/C, Rh/C, PdRhNi/C and Pd_4_Rh_2_Ni_1_/C. The consistent 25° broad peak is attributed to the graphitic Vulcan carbon. The Pd/C pattern shows four peaks of Pd (111), (200), (220) and (311) at approximately 40°, 46°, 68° and 82.5°, respectively (JCPDS card, File No. 46-1043). The Rh/C pattern shows only the Rh (111) facet at 41.5°, while other peaks are absent (Rh PDF#05-685). The Ni one shows peaks of Ni(OH)_2_ at 35° and 60° and a peak of NiO at 44.5°, which were also reported in [42]. It is noteworthy that the C peak is more intense than the Rh and Ni ones on Rh/C and Ni/C, respectively, which signifies a weak crystallinity of both Rh and Ni. In this context, it is noteworthy that, in previous works, replacing Rh with Au or Ag in PdAuNi/C and PdAgNi/C has enhanced the Pd EOR activity [33,43]. Unlike Rh, both Au and Ag possess larger crystal lattices than Pd, and therefore their diffraction angles are smaller than those of Pd. That is why, while adding Ag or Au to Pd exercises a tensile strain on the Pd lattice, adding Rh exercises compressive strain. This is demonstrated by the XRD patterns of Pd/C, Ni/C, Rh/C, PdRhNi/C and Pd_4_Rh_2_Ni_1_/C. The consistent 25° broad peak is attributed to the graphitic Vulcan carbon. The Pd/C pattern shows four peaks of Pd (111), (200), (220) and (311) at approximately 40°, 46°, 68° and 82.5°, respectively (JCPDS card, File No. 46-1043). The Rh/C pattern shows only the Rh (111) facet at 41.5°, while other peaks are absent (Rh PDF#05-685). The Ni one shows peaks of Ni(OH)_2_ at 35° and 60° and a peak of NiO at 44.5°. It is noteworthy that the C peak is more intense than the Rh and Ni ones on Rh/C and Ni/C, respectively, which signifies a weak crystallinity of both Rh and Ni. In this context, it is noteworthy that in previous works, replacing Rh with Au or Ag in PdAuNi/C and PdAgNi/C enhanced the Pd EOR activity [33,43]. This can be elucidative regarding the impact of adding Rh to Pd, which, unlike Au and Ag, shifts the Pd XRD diffraction angles to higher values. Thus, Au and Ag shifts Pd peaks to lower values and exercises tensile lattice strain. However, Rh shifts the Pd peaks to higher angles and exercises compressive lattice strain.

The trimetallic sample patterns demonstrate single phase peaks of (111), (200), (220) and (311) facets that are located between the monometallic Pd and the Rh one. The peak shift to higher diffraction angles is suggestive of alloy formation between Pd and Rh. Furthermore, the trimetallic patterns do not show single Ni or Ni oxide peaks, which implies the incorporation of Ni species into the Pd lattice. Moreover, the trimetallic peaks are generally broader than the monometallic Pd/C ones. That is probably due to the smaller crystal size of trimetallic samples. The crystal size (*τ*, nm) is estimated using Scherrer’s Equation (1):(1)$$\tau =\frac{K\lambda }{\beta \cos\left(\theta \right)}$$
where *K* is a constant and equals 0.94, *λ* is the wavelength (0.154 nm for Cu kα), *β* is the full width at half maximum of the peak in radians and *θ* is the half diffraction angle. Utilizing the (111) peak details, the crystallite sizes of Pd, Pd_4_Rh_2_Ni_1_ and PdRhNi are found to be 4 nm, 3.77 nm and 2.46 nm, respectively. It is noteworthy that adding Rh and Ni in Pd_4_Rh_2_Ni_1_ decreased the particle size and that further increasing the Rh and Ni content in PdRhNi/C further decreases the crystallite size. Figure 4 shows the TEM micrographs and particle size distribution of Pd/C (Figure 4A,B), Pd_4_Rh_2_Ni_1_/C (Figure 4C,D) and PdRhNi/C (Figure 4E,F). Highly dispersed spherical metal nanoparticles could be seen in the three micrographs on the carbon aggregates (30–60 nm in size). The highest dispersion was that of PdRhNi/C, followed by Pd_4_Rh_2_Ni_1_/C, and finally Pd/C. Each sample micrograph was examined, and 100–200 particles were manually selected to estimate the particle size distribution. The particle size distribution graphs demonstrate that the smallest particle size was that of PdRhNi with an average of 3 nm, followed by Pd_4_Rh_2_Ni_1_ with an average of 3.6 nm and finally Pd with an average of 5 nm. This is consistent with the XRD crystal size and peak broadening (Figure 3) trend, which is suggestive of the alloying effect among Pd, Rh and Ni.

Table 2 lists the measured chemical composition and metal oxide percentage measured by XPS and EDX techniques. The variation between XPS and DEX concentration values is probably due to the depth difference that each technique penetrates at the sample. Moreover, different metal surface segregation is expected. It is noteworthy that on PdRhNi—where the nominal Pd:Rh:Ni atomic ratio is 1:1:1—the XPS concentrations were usually more than the EDX ones due to the metal nanoparticles’ tendency of surface segregation and XPS surface sensitivity. The Pd atomic concentration was highest on the monometallic Pd/C and lowest on PdRhNi/C. The opposite trend was noted for Rh and Ni by shifting from Pd_4_Rh_2_Ni_1_ to PdRhNi. Chemical composition (oxide concentration) was measured by XPS and EDX (10 kV) for Pd/C, Pd_4_Rh_2_Ni_1_/C and PdRhNi/C.

Figure 5 demonstrates the XPS spectral peaks of Pd/C, Pd_4_Rh_2_Ni_1_/C and PdRhNi/C. The Pd 3d spectrum of Pd/C (Figure 5A) shows an energy doublet of lower-energy Pd 3d_5/2_ band and higher-energy Pd_3/2_. Additionally, the Pd oxide’s peaks are visible, which indicates the poor Pd air stability. However, the addition of Rh and Ni enhanced the air stability even though it did not completely prevent PdO formation, as shown in Figure 5C,F. Most of the Ni in both Pd_4_Rh_2_Ni_1_/C and PdRhNi/C exist in an oxidized state, which is substantiated by the XRD patterns (Figure 3). Twenty percent of the Rh in PdRhNi/C exists in an oxidized state, while the Pd_4_Rh_2_Ni_1_/C contains only metallic Rh. Pd oxide formation in the PdRh/C sample has also been reported [25]. Figure 5B shows the Pd 3d peak shifted to higher energy in the case of the trimetallic catalysts compared with the monometallic ones. Table 1 demonstrates the shift is 0.11 eV and 0.06 eV for Pd_4_Rh_2_Ni_1_/C and PdRhNi/C, respectively. This binding energy shift is likely to have impacted the adsorption characteristics of Pd with ethanol and OH species, which might help explain the low catalytic activity of the trimetallic catalysts.

Moreover, when EDX was performed at 20 kV (i.e., larger surface depth), the Rh and Pd concentrations increased 0.23 at.% and 0.08%, respectively, but the Ni one decreased 0.12 at.%. This could be explained by the catalyst surface being overall rich in Rh, while Ni tends to segregate into the higher surface layers. More interestingly, Table 2 lists the measured chemical composition and metal oxide percentage measured by XPS and EDX techniques. The variation between XPS and DEX concentration values is probably due to the depth difference that each technique penetrates at the sample. Moreover, different metal surface segregation is expected. It is noteworthy that on PdRhNi—where the nominal Pd:Rh:Ni atomic ratio is 1:1:1—the XPS concentrations were usually more than the EDX ones due to the metal nanoparticles’ tendency of surface segregation and XPS surface sensitivity. The Pd atomic concentration was highest on the monometallic Pd/C and lowest on PdRhNi/C. The opposite trend was noted for Rh and Ni by shifting from Pd_4_Rh_2_Ni_1_ to PdRhNi. Figure 5 demonstrates that the Pd 3d_5/2_ binding energy of Pd/C was located at 335.43 eV. However, it shifted to higher values for the trimetallic samples: 0.11 eV and 0.06 eV for Pd_4_Rh_2_Ni_1_/C and PdRhNi/C, respectively. This could be explained by the adsorption characteristics of Pd changing in the trimetallic samples and adsorbing the EOR intermediates more strongly than necessary. Consequently, many surface Pd active sites on PdRhNi/C samples are easily poisoned and blocked from the electroactive species.

## 4. Conclusions

Two samples of trimetallic PdRhNi/C nanoparticles, along with their monometallic Pd/C, Rh/C and Ni/C counterparts, were successfully prepared via a scalable borohydride reduction route. Neither Rh or Ni present sensible activity toward EOR, unlike Pd. Unfortunately, the trimetallic samples underperformed their monometallic Pd/C counterpart on EOR. Although adding Ni or Rh to Pd/C is extensively reported to enhance its EOR activity, their simultaneous addition using the current method gave the opposite outcome. The exact solid reasons for such lower performance are unknown, but some other findings could be drawn up. Neither Rh/C nor Ni/C could activate EOR at the current potential window. Their addition seems to have reduced the Pd surface coverage, reducing the EOR active surface area on the trimetallic samples. Moreover, the Ni and Rh addition shifted the Pd 3d binding energy to higher values, increasing their adsorption strength of EOR intermediates and reducing their CO-like species tolerance.

## Figures and Tables

**Figure 1 micromachines-14-00957-f001:**
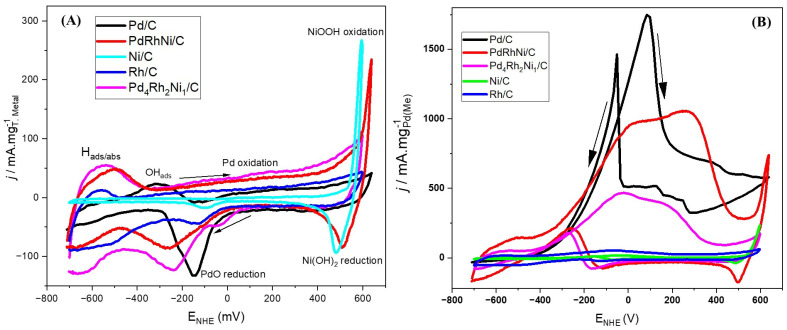
Cyclic voltammetry (CV) scans in 1 M KOH (**A**) and 1 M KOH+ C_2_H_5_OH (**B**) of Pd/C, Rh/C, Ni/C, PdRhNi/C, Pd_4_Rh_2_Ni_1_/C at 50 mV/s.

**Figure 2 micromachines-14-00957-f002:**
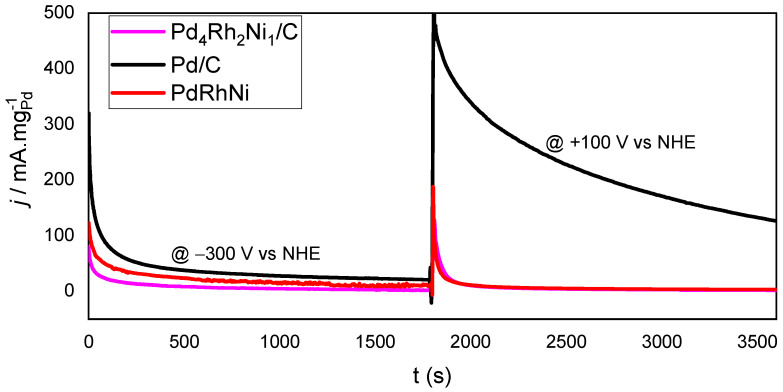
Two-step CA scans of Pd/C, PdRhNi/C and Pd_4_Rh_2_Ni_1_/C at −300 V and +100 V vs. NHE in 1 M KOH + C_2_H_5_OH.

**Figure 3 micromachines-14-00957-f003:**
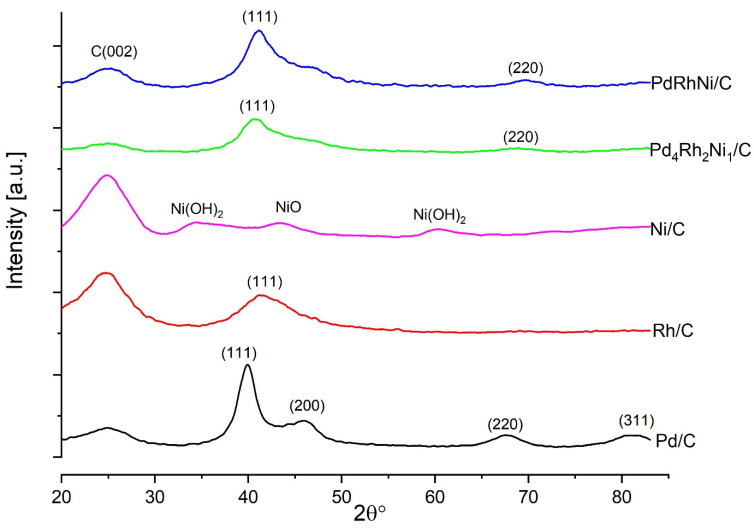
X-ray diffraction (XRD) patterns of Pd/C, Rh/C, Ni/c, PdRhNi/C and Pd_4_Rh_2_Ni_1_/C.

**Figure 4 micromachines-14-00957-f004:**
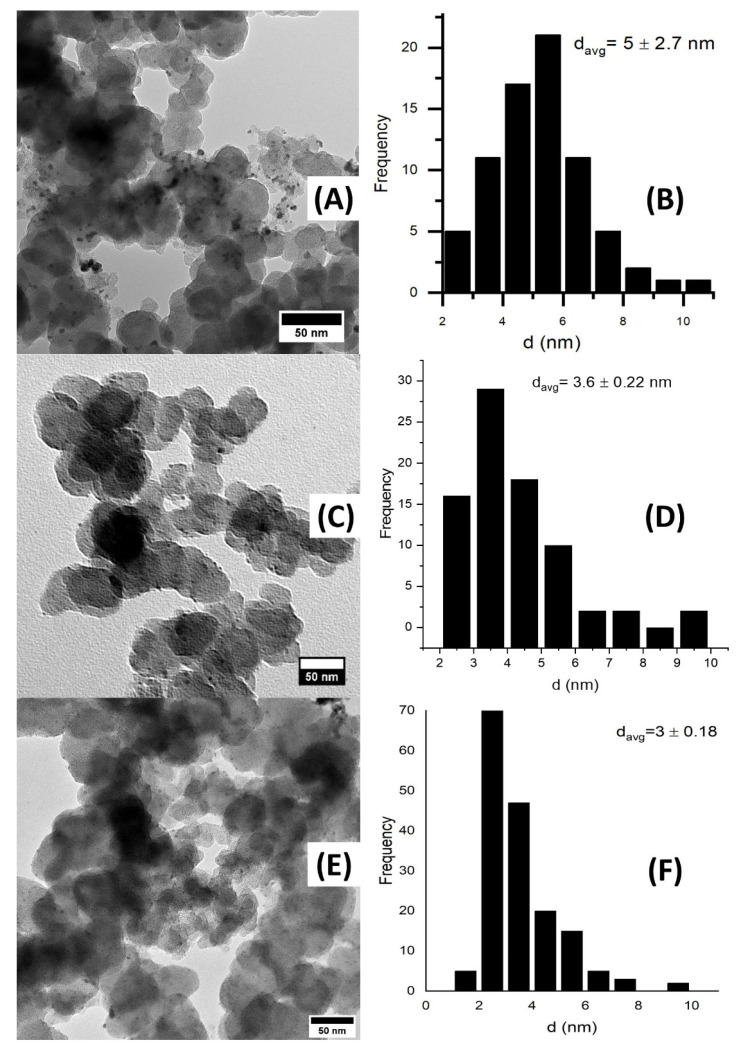
TEM Micrographs and particle size distribution of Pd/C (**A**,**B**), Pd_4_Rh_2_Ni_1_/C (**C**,**D**) and PdRhNi/C (**E**,**F**).

**Figure 5 micromachines-14-00957-f005:**
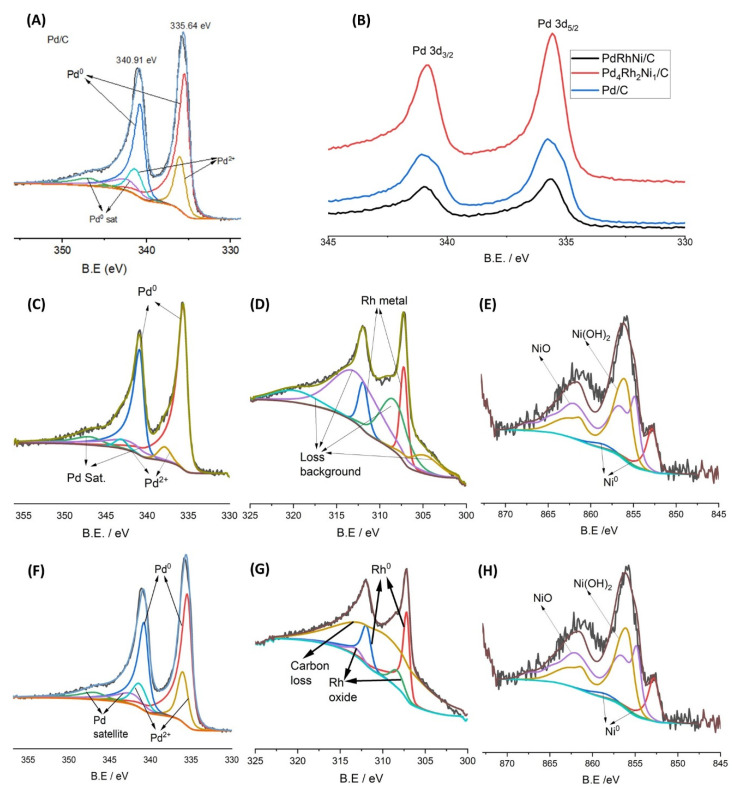
XPS spectra of Pd 3d of Pd/C (**A**), Pd 3d band comparison (**B**) and elemental peaks of Pd_4_Rh_2_Ni_1_/C (**C**–**E**) and PdRhNi/C (**F**–**H**).

**Table 1 micromachines-14-00957-t001:** Stoichiometric quantities of C, PdCl_2_, NiCl_2_ and RhCl_3_ to prepare Pd/C, Ni/C, Rh/C, Pd_4_Rh_2_Ni_1_/C and PdRhNi/C.

	C (mg)	PdCl_2_ (mg)	NiCl_2_ (mg)	RhCl_3_ (mg)
Pd/C	132	30	-	-
Rh/C	132	-	-	36
Ni/C	132	-	22	-
PdRhNi/C	132	11.91	8.67	11.51
Pd_4_Rh_2_Ni_1_/C	132	18.40	3.4	10.89

**Table 2 micromachines-14-00957-t002:** EDX and XPS elemental concentration measurements of Pd/C, Pd_4_Rh_2_Ni_1_/C and PdRhNi/C.

Catalyst	Pd at.%		Rh at.%		Ni at.%		Notes (XPS)	Pd 3d_5/2_ (eV)
	XPS	EDX	XPS	EDX	XPS	EDX		
Pd/C	2.1	-	-	-	-	-	0.45% Pd^2+^	335.43
Pd_4_Rh_2_Ni_1_/C	1.7	0.95	0.4	0.68	0.20	0.25	0.08% Pd^2+^ 0.14% Ni^2+^	335.54
PdRhNi/C	1.2	0.39	0.70	0.43	0.70	0.50	0.06% Ni^0^ 0.09% Pd^2+^ 0.15% Rh^2+^	335.49

## Data Availability

The raw research data of this work are available at DOI: 10.17632/hp22gbrn9d.1.

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
