# Peer review of "Unexpected Negative Performance of PdRhNi Electrocatalysts toward Ethanol Oxidation Reaction"

_micromachines, 2023, doi:10.3390/mi14050957_

Round 1

Reviewer 1 Report

This work investigates the trimetallic (Pd-Rh-Ni) Carbon supported catalysts for alkaline ethanol oxidation. The authors have done a good amount of physical and electrochemical characterization. However, they failed to present the data well. I do not recommend the article in its current form. A few of my concerns are given below;

1.       The abstract is too simple and not informative. It should be described highlighting the importance of this work.

2.       Page 3, line 116, “The obtained current is divided by the total metal load in the respective catalyst.” What is the mass loading used for the electrochemical analysis?

3.       The representation of the data is too poor (Figures).

4.       Figure 3 is very confusing. The authors should carry out baseline correction and appropriately plot the graphs. The peaks are just assigned without any JCPDS Number or any related reference.

5.       In Figure 4, all the bare scales are represented differently. There is no uniformity throughout the manuscript. All the micrographs look alike. Where are the frequency values for Figures 4 B and F?

6.       The manuscript needs careful re-reading and language polishing 

Author Response

Dear Reviewer, 

Thank you for taking the time to review our work. 

We received your comments and have tried to adopt your advise and fix the problems you highlighted.

Please find our response in the attachment. 

Best wishes,

The authors 

Reviewer 2 Report

The article "Unexpected negative performance of PdRhNi electrocatalysts toward ethanol oxidation reaction" presents a complete study. The work is of absolute interest from a scientific point of view. However, there are the following comments and questions:

1.      Figure 4: How was the particle size distribution obtained? Is it possible to add high magnification micrographs?

2.      Lines 116, 150, 170, 188, 190 and further down the text – Pd4Rh2Ni1/C - number font should be subscript.

3.      Line 127, Line 131 – Hads/abs subscript font should be used.

4.      Line 135 – There is no (b) in the figure caption.

5.      Line 148 – The second peak is located at 200-300 mV rather than at 400 mV.

6.      Line 161 – “… sample is more tolerant than CO poisoning species.” – the phrase is not clear.

7.      Figure 2 – Why is the initial value of the current density at -300mV for the Pd/C sample higher than for the trimetallic samples but all these samples have the same values at the Figure 1B?

8.      Line 175 - Ni(OH)2 - number font should be subscript

9.      Figure 3 -Peaks corresponding to Rh/C should be marked. Curve for the Ni/C should be located higher and not cross other lines. Vertical lines indicated peaks should be added.

Author Response

Dear Reviewer,

Thank you for taking the time to review our work. We appreciate your time and effort to improve and better communicate our article. 

We have adopted your comments and fixed the ambiguities you have highlighted in the manuscript. 

Please find our response in attachements. 

We look forward to your feedback. 

Best wishes,

Ahmed 

Round 2

Reviewer 1 Report

The authors have made the necessary changes, and the manuscript can be accepted for publication.